# Lightning UQ Box: A Comprehensive Framework for Uncertainty Quantification in Deep Learning

## Abstract

Uncertainty quantification (UQ) is an essential tool for applying deep neural networks (DNNs) to real world tasks, as it attaches a degree of confidence to DNN outputs. However, despite its benefits, UQ is often left out of the standard DNN workflow due to the additional technical knowledge required to apply and evaluate existing UQ procedures. Hence there is a need for a comprehensive toolbox that allows the user to integrate UQ into their modelling workflow, without significant overhead. We introduce `Lightning UQ Box`: a unified interface for applying and evaluating various approaches to UQ. In this paper, we provide a theoretical and quantitative comparison of the wide range of state-of-the-art UQ methods implemented in our toolbox. We focus on two challenging vision tasks: (i) estimating tropical cyclone wind speeds from infrared satellite imagery and (ii) estimating the power output of solar panels from RGB images of the sky. By highlighting the differences between methods our results demonstrate the need for a broad and approachable experimental framework for UQ, that can be used for benchmarking UQ methods. The toolbox, example implementations, and further information are available at: https://github.com/lightning-uq-box/lightning-uq-box.

## 1 Introduction

In real world applications, deep learning (DL) models are often deployed in safety-critical domains such as healthcare [45], robotics [56], and Earth observation [55, 60], with relevant areas including flood monitoring [7], wildfire mapping and forecasting [58], and weather forecasting [59]. In these fields, an incorrect prediction can cause significant damage and corresponding consequences. Uncertainty quantification (UQ) aims to provide a measure of confidence about a neural network's prediction and to support practitioners in identifying potentially false predictions to better guide analyses and decision-making processes [21]. Besides this, UQ can even improve predictive performance via regularization [16, 39].

The direct application of UQ to DL is often not straightforward for practitioners. Besides the implementation challenges associated with probabilistic modelling and stochastic training algorithms, the performance of UQ methods can fluctuate, depending on the data and the task [50]. Moreover, there is a lack of clear guidance on which methods are promising for specific tasks, given the ever-increasing zoo of UQ methods for DL [1, 21]. These challenges are particularly prominent for data modalities of higher dimensions, such as vision, where uncertainty modelling adds a further layer of complexity. Therefore, various approaches need to be considered, which usually involve different loss functions, training procedures, and model architectures. The need of accessible and open-source UQ frameworks is also called upon in a recent position paper on Bayesian Deep Learning (BDL) by leading experts in this field [53]: "Software development is key to encouraging DL practitioners to use Bayesian methods. More generally, there is a need for software that would make it easier for

practitioners to try BDL in their projects. The use of BDL must become competitive in human effort with standard deep learning." [53].

`Lightning UQ Box` provides users with all the tools needed to equip deep neural networks (DNNs) with UQ. We created `Lightning UQ Box` to tackle the gap between theoretical researchers and actual practitioners in the field of UQ in DL. The toolbox offers a comprehensive framework, building on top of `PyTorch` [54] and `Lightning` [18], as an accessible end-to-end solution. The toolbox is particularly suited for vision applications (see Section 3): it offers flexible layer configurations like Bayesian convolution layers that can be modularly placed in backbone architectures, which streamline UQ.

We underline the usefulness of the presented toolbox with two example applications: estimating the maximum sustained wind speed of tropical cyclones from satellite imagery and predicting the power voltage output of solar panels from a time series of sky images. These applications contain different sources and types of uncertainties in the input and target variables and illustrate the stochastic nature of real world phenomena and measurement systems practitioners are confronted with. Simultaneously, these applications carry an associated inherent risk that demands reliable predictive uncertainties. The central contributions of our work all aim to equip practitioners with the necessary tools to apply UQ methods for DL on their specific (real world) problem:

- **Comprehensive End-to-End UQ Toolbox:** `Lightning UQ Box` enables practitioners to efficiently iterate over ideas without having to re-implement the provided UQ methods. To do so, it provides backbone architecture- and dataset-agnostic implementations of a wide array of UQ methods and corresponding evaluation schemes for DL, covering regression, classification, semantic segmentation, and pixel-wise regression tasks.

- **Adaptability and Expandability:** The modular implementation using `Lightning` encourages practitioners and the community to an individual adaptation and a continuous expansion and growth of the toolbox. Additionally, the implementation is adapted to vector or vision data. Specifically, partial stochasticity [65] is supported when applicable. This supports any larger architectures used for vision, and the "frozen" functionality enables retraining only a few layers.

- **Practical and Theoretical Introductions:** The toolbox contains comprehensive practical and theoretical introductions to the field of UQ and the application of the toolbox. UQ Tutorials and case studies on designing downstream tasks to compare various UQ methods are provided. A comprehensive theory guide provides methodological backgrounds on the implemented methods.

**Related Work** Frameworks for UQ in DL already exist in the `PyTorch` [54] ecosystem. However, they are limited to either a handful of UQ methods or a specific class of approaches, such as BDL. Several libraries exist for BDL, most notably `TorchBNN` [38], `BLiTZ` [17], and `Bayesian-Torch` [35]. Yet BNNs are only one approach to UQ and require choosing a prior distribution. When an abundance of data is available, frequentist procedures, such as conformal prediction, can be a more attractive alternative. The library `Fortuna` [14] supports several approaches to conformal prediction (CP), of which we currently support a subset (with plans to incorporate more). The primary difference between our work and `Fortuna` is that `Fortuna` is primarily compatible with `JAX` [9] and only supports post-hoc calibration of `PyTorch` models. `TorchCP` [71] is another framework that implements conformal prediction [4], but it does not support other UQ methods (such as BDL). The most closely related package to ours is `torch-uncertainty` [36], which implements both frequentist and Bayesian UQ methods in addition to common benchmarks. Yet our `Lightning UQ Box`, to date, implements the largest number of UQ methods across different theoretical frameworks, such as BDL and CP, while including cutting-edge techniques as partially stochastic networks [65], and additionally supports UQ methods for semantic segmentation tasks. Table 1 gives a comparison with previous libraries.

## 2  Benchmarking UQ Methods: the Lightning UQ Box

The underlying design of `Lightning UQ Box` is based on three pillars:

- provide a comprehensive set of reference implementations of state-of-the-art UQ methods,

- optimally fit in the wide open-source landscape for DL based on `PyTorch`, and

- enhance automation, scalability, and reproducibility of experiments with `Lightning`.

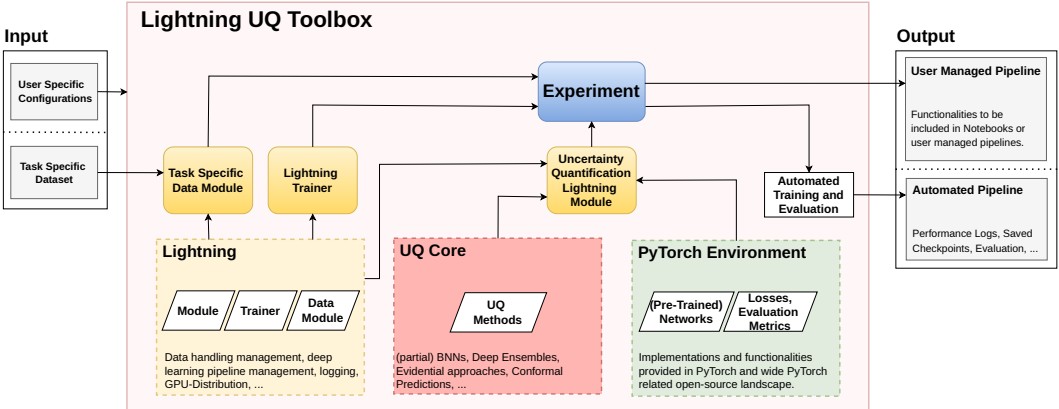

Figure 1: The structure of `Lightning UQ Box`. The experiments can be built and evaluated at scale or manually tailored to specific use cases. For large experiments at scale, only a dataset and a configuration file have to be provided.

These design goals are reflected in the structure of the toolbox, as visualized in Figure 1, and build up on the three core components of the available DL functionalities provided within the `Lightning` framework for structuring and pipeline managing, the UQ Core, which contains the UQ method implementations, and the `PyTorch` ecosystem.

The UQ Core contains a comprehensive collection of UQ methods for DL with different theoretical underpinnings consolidated and implemented for this toolbox. The theoretical backgrounds are very diverse and cover, for example, mean-field estimation and various Bayesian-motivated approaches, including kernel-based approaches and partially stochastic networks, ensemble methods, and evidence-motivated approaches (cmp. Section 2.1). Besides the diversity in methodological approaches, the toolbox provides unified interfaces and configuration patterns, thereby improving accessibility and, importantly, enabling comparability between the methods.

The toolbox is compatible with common DL libraries and frameworks from the `PyTorch` ecosystem. The provided UQ methods can be combined with user-specific architectures and implementations provided in the `PyTorch` ecosystem, including pre-trained networks and foundation models. This is especially useful as our framework can build upon or be included in existing code and pipelines based on `PyTorch`-based libraries, such as `timm` [72]. In order to scale BDL to modern-sized architectures, we offer functionality to convert existing deterministic architectures, or specified components thereof, automatically to a Bayesian framework. As a result, the collection of UQ methods goes beyond mere method compilation, offering not only comprehensiveness but also removing time-consuming implementation overhead. This enables users to use the UQ toolbox as a simple extension of their existing DL pipelines.

The toolbox utilizes the `Lightning` framework to enhance experiment automation, scalability, and reproducibility. `Lightning` offers a flexible and user-friendly interface for the automated management of complex pipelines. It is specifically designed to support practitioners in managing experiments by providing functionalities to enhance their scalability and reproducibility. These include managing configurations, training loops, evaluation steps, and logging processes. To this end, each UQ method is implemented as a `LightningModule` that can be used with a `LightningDataModule` and a `Trainer` to execute training, evaluation, and inference for a desired task. The toolbox also utilizes the `Lightning` command line interface (CLI) for better experiment reproducibility and for setting up experiments at scale. This provides an end-to-end configuration, such that a full pipeline of experiments can be built with minimal overhead. Many optional configurations and user-specific objects, such as logging functionalities or models, can be included but are not mandatory. The general concept of the toolbox is illustrated in Figure 1.

## 2.1 Provided Types of UQ Methods

`Lightning UQ Box` provides the most comprehensive collection of the extensive and versatile landscape of UQ methods for DL. The following section gives an overview of these different UQ

Table 1: The methods provided with `Lightning UQ Box` and other available frameworks and reviews. The table represents the status at the time of publication and will be extended in the future. All currently available methods can be found in the provided repository.

| Publication | [26] | [63] | [15] | [31] | [64] | [50] | [46] | [36] | Lightning UQ Box |
|---|---|---|---|---|---|---|---|---|---|
| **Deterministic Methods** | | | | | | | | | |
| Gaussian (MVE) | ✓ | | | | | | ✓ | | ✓ |
| Deep Evidential Networks (DER) | | | | | | | ✓ | | ✓ |
| **Neural Network Ensembles** | ✓ | ✓ | ✓ | ✓ | ✓ | ✓ | ✓ | ✓ | ✓ |
| **Bayesian Neural Networks** | | | | | | | | | |
| MC Dropout (GMM) | | ✓ | ✓ | ✓ | ✓ | ✓ | ✓ | ✓ | ✓ |
| BNN with VI ELBO | | | | ✓ | ✓ | ✓ | ✓ | | ✓ |
| BNN with VI (alpha divergence) | | | | | | | | | ✓ |
| VBLL | | | | | | | | | ✓ |
| Laplace Approximation | | | | | ✓ | | | | ✓ |
| SWAG | | | | ✓ | ✓ | | | | ✓ |
| DVI, SI | | | | ✓ | | | | | |
| HMC | | | | ✓ | | | | | |
| Radial BNN | | | | | | ✓ | | | |
| Rank-1 BNN | | | | | | ✓ | | | |
| **Gaussian Process based** | | | | | | | | | |
| Deep Kernel Learning (DKL) | | | | | | | | | ✓ |
| Det. Unc. Estimation (DUE) | | | | | | | | | ✓ |
| Spectral Normalized GPs (SNGP) | | | | | ✓ | ✓ | | | ✓ |
| **Quantile based** | | | | | | | | | |
| Quantile Regression (QR) | ✓ | | ✓ | | | | | | ✓ |
| Conformal Prediction (CQR) | ✓ | | ✓ | | | | | | ✓ |
| **Diffusion Model** | | | | | | | | | |
| CARD | | | | | | | | | ✓ |
| **Post-hoc Calibration** | | | | | | | | | |
| RAPS | | | | | | | | | ✓ |
| TempScaling | | | | | | ✓ | | ✓ | ✓ |

methods, which are listed in Table 1. For comprehensive explanations, we refer to the theory guide in the supplement and to existing reviews [1, 21]. For **regression tasks** NNs predict a continuous target $y^\star$. Currently, the toolbox supports six classes of UQ methods for regression: deterministic, quantile, ensemble, Bayesian, Gaussian Process, and diffusion-based methods.

1. Deterministic methods: use a DNN, $f_\theta : X \rightarrow \mathcal{P}(Y)$, that map inputs $x$ to the parameters of a probability distribution $f_\theta(x^\star) = p_\theta(x^\star) \in \mathcal{P}(Y)$, and include methods like Deep Evidential Regression (**DER**) [2] and Mean Variance Estimation (**MVE**) [49]. The latter, for example, outputs the mean and standard deviation of a Gaussian distribution $f_\theta^{\mathrm{MVE}}(x^\star) = (\mu_\theta(x^\star), \sigma_\theta(x^\star))$.

2. Quantile based models: use a DNN, $f_\theta : X \rightarrow Y^n$, that map to $n$ quantiles, $f_\theta(x^\star) = (q_1(x^\star), ..., q_n(x^\star)) \in Y^n$, and include Quantile Regression [33] (**Quantile Regression**) and the conformalized version thereof (**ConformalQR**) [62].

3. Ensembles: Deep Ensembles [37], which utilize an ensemble over MVE networks.

4. Bayesian methods: model the network parameters as random variables. Multiple principles and techniques to approximate BNNs have been introduced. We include BNNs with Variational Inference (VI) (**BNN VI ELBO**) [8], BNNs with VI and alpha divergence (**BNN VI**) [13], Variational Bayesian Last Layers (**VBLL**) [28], MC-Dropout (**MCDropout**) [20], the Laplace Approximation (**Laplace**) [61][12] and **SWAG** [43] with partially stochastic variants [65].

5. Gaussian Process-based methods: these model a joint distribution over a set of functions in a data-driven manner that approximates the first and second moment of the marginalized distribution. These include Deep Kernel Learning (**DKL**) [73], an extension thereof Deterministic Uncertainty Estimation (**DUE**) [69, 70], and Spectral Normalized Gaussian Process (**SNGP**) [40].

6. Conditional Generative model: Classification and Regression Diffusion (**CARD**) [27].

For **classification**, the toolbox currently supports six classes of UQ methods. Vanilla softmax probabilities can be directly used to obtain predictive uncertainties. However, they are often miscalibrated and have lead to post-hoc recalibration methods being proposed [25].

1. Deep Ensembles (**DeepEnsembles**) [37]: utilize an ensemble over independent standard classification networks.

2. Bayesian methods: **BNN VI ELBO** [8], **VBLL** [28], **MCDropout** [20], **Laplace** [61][12], **SWAG** [43].

3. Gaussian Process based methods: **DKL** [73], **DUE** [69] and Spectral-normalized Neural Gaussian Processes (**SNGP**) [40].

4. Conformal Prediction: [62], Regularized Adaptive Prediction Sets (**RAPS**) [3].

5. Other: Test-time Augmentation (**TTA**) [41], Temperature Scaling [25].

Additionally to the general purpose tasks of regression and classification, `Lightning UQ Box` supports UQ methods for vision-specific tasks. These include segmentation and pixel-wise regression, where an extensive overview of supported UQ methods can be found on our documentation page.

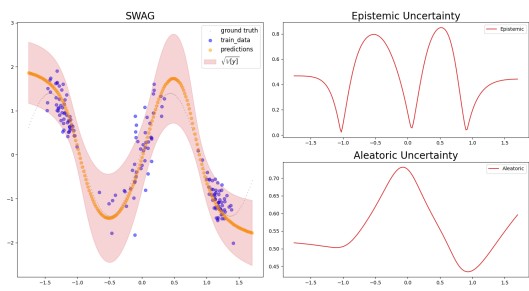

```
1   dm = ToyDataModule()
2   swag_model = SWAGRegression(
3       deterministic_model.model,
4       num_mc_samples=50,
5       swag_lr=1e-3,
6       loss_fn=NLL(),
7   )
8   swag_trainer = Trainer(
9       max_epochs=20,
10  )
11  swag_trainer.fit(swag_model, datamodule=dm)
12  preds = swag_model.predict_step(X_test)
```

(a) Example code to fit SWAG method.     (b) SWAG regression toy example.

Figure 2: Example code and visualization on toy regression dataset.

**Quantifying Predictive Uncertainty:** By default, we quantify predictive uncertainty via the standard deviation for regression and via the entropy of the predictive distribution for classification. In general, for UQ in DL, two main types of uncertainties can be considered: aleatoric and epistemic [13, 21]. Aleatoric uncertainty originates from random, or partially observable, effects in the data itself and is not reducible: for instance, the Earth covered with clouds does not contain enough information to surely assign the land cover type to one of multiple given options. In contrast, epistemic uncertainty quantifies the model's predictive uncertainty originating from uncertainty over its parameters: it will typically shrink as more data becomes available [30]. See Figure 2b for an example decomposition. Depending on the underlying theoretical assumptions, UQ methods model these types of uncertainties individually or mutually [30]. From a statistical perspective, Gruber et al. [24] allude that such a distinction is often not possible. Thus, in the examples given here, we focus on the approximate predictive distributions of the UQ methods $p_\theta(y_\star|x_\star)$, from which we derive the aforementioned uncertainty measures. However, where applicable, the toolbox also enables researchers to decompose these two types of uncertainties.

**Limitations:** Despite the robustness and versatility of the `Lightning UQ Box`, it is tightly integrated within the `PyTorch` ecosystem, limiting its applicability to other existing DL frameworks like `Tensorflow` and `JAX`. Furthermore, merely using UQ methods does not guarantee complete reliability, and applications nevertheless require proper experimental design and evaluation.

## 3 Experimental Setup for Validation

We now showcase `Lightning UQ Box` as a valuable tool for conducting experimental studies including benchmarking. We exemplify this by comparing UQ methods on three challenging computer vision datasets from two different domains. More concretely, we evaluate the methods on selected downstream tasks that highlight the efficacy of UQ and the usefulness of a unified framework[1]. Each experiment was completed using the UQ toolbox in less than 10 hours (6 hours on average) on a single A100 40GB GPU.

---

[1]Code for all experiments available at this link: Github Repo.

## 3.1 Datasets

For our experiments, we consider three datasets: the Tropical Cyclone Driven Data Challenge dataset (TC) [44], the Digital Typhoon (DT) dataset [32], and the SKy Images and Photovoltaic Power Generation Dataset (SKIPP'D) [47]. An overview of the datasets is given in Table 2. For a detailed explanations of the datasets see supplementary section 1.

Table 2: Dataset Overview.

| Dataset | Image source/Satellite | Spatial Res. | Temporal Res. | Train Samples | Val. Samples | Test Samples |
|---|---|---|---|---|---|---|
| Tropical Cyclone | GOES | 2 km | 15 min | 53k | 11k | 43k |
| Digital Typhoon | Himawari | 5 km | 60 min | 64.5k | 14k | 20k |
| SKIPP'D | Fisheye camera | - | 1 min | 280k | 63k | 14k |

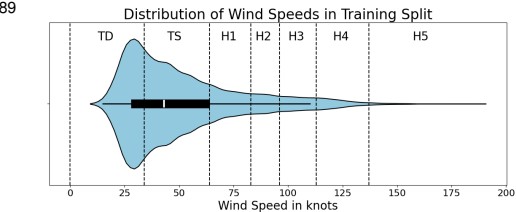

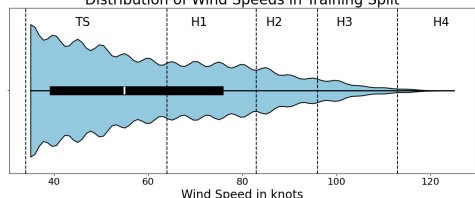

(a) Label distribution and storm categories.

(b) Label distribution and storm categories.

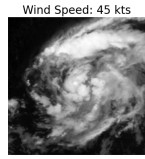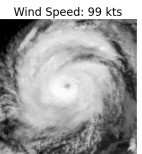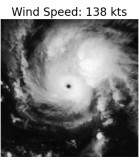

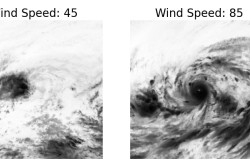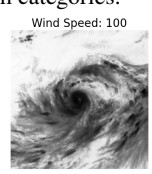

(c) Samples from the Tropical Cyclon Dataset.

(d) Samples from the Digital Typhoon Dataset.

Figure 3: Visualization of the Tropical Cyclon (left) and the Digital Typhoon Dataset (right).

**Cyclone and Typhoon Dataset:** The TC and DT datasets consist of infrared measurements that capture the spatial structure of storms. Corresponding wind speed targets are matched based on hurricane databases. There are varying sources of uncertainty in the inputs, such as missing pixels due to the swath of the satellites, and in the targets, such as measurement uncertainties and interpolations over time with respect to non-uniform time steps. As such, these datasets exemplify real world stochastic phenomena, where predictive uncertainties are essential for decision-making due to the inherent risk associated with these potentially extreme events. The magnitude of rapid intensification events has been increasing [6], thus causing more damage if not properly detected and predicted. One such recent example is Hurricane Otis in October 2023, where existing models had to disproportionally rely on satellite data, due to limited in-situ data, which lead to erroneous forecasts [34]. Given the extensive availability of satellite imagery, research efforts using this modality are a promising avenue to enhance existing forecasts.

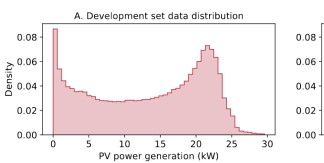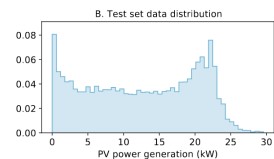

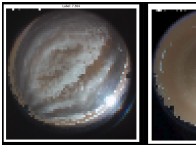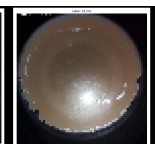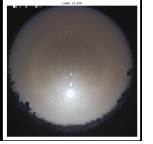

(a) Statistics of SKIPP'D test and train set [47].

(b) Example Image of the SKIPP'D dataset.

Figure 4: Visualization of SKIPP'D Dataset.

**Photovoltaic Dataset:** The SKIPP'D dataset consists of ground-based fish-eye RGB images over a 3-year period (2017–2019), where associated targets are power output measurements from a 30 kilowatt (kW) rooftop photovoltaic array [47]. Given the urgent necessity to transform the world's energy sector to more sustainable solutions [5], this dataset aims to support research efforts of large-scale integration of power voltage into electricity grids, where the main problem is to manage the non-constant and intermittent power source [47].

 **3.2 Methodological Setup**

**Cyclone and Typhoon Dataset:** Various works have framed the wind speed estimation of tropical cyclones from a satellite image as both a regression [10, 42, 75] and classification [57, 74] task. We apply all UQ methods provided by the toolbox to the regression and classification task. For all wind speed experiments, we use the same ResNet-18 [29] pre-trained on ImageNet[2] as the backbone architecture of compared UQ methods. For the TC and DT datasets, the chosen task is selective prediction, as introduced by Geifman et al. [22]. Here, samples with a predictive uncertainty above a given threshold are omitted and can be referred to domain experts or further decision-making pipelines. If the corresponding UQ method has higher uncertainties for inaccurate predictions, leaving out the predictions for these samples should increase the overall accuracy, indicating a correlation between predictive uncertainty and model error. This can be beneficial in a deployment setting where automated analysis systems are paired with human expertise. Examples are visualized in Figure 6.

**Photovoltaic Dataset:** Previous work have demonstrated promising results of such image data for photovoltaic power generation estimation modeled as a regression task [67, 76, 68, 48, 19, 51, 52]. We apply all UQ methods provided by the toolbox (see Table 1) to this regression task. Here, we use the proposed CNN architecture of Nie et al. [47], which requires only a single line code change in experiment configuration for each respective UQ method.[3] Given the central problem of photovoltaics being a non-constant power source, we analyze the additional benefits of UQ by evaluating predictive uncertainty on annotated sunny and cloudy days. From a reliable model, we expect that both the predictive error as well as the predictive uncertainty is larger on the cloudy samples because the partial occlusions make it more difficult to estimate the corresponding power voltage output.

**Evaluation Metrics:** As evaluation metrics, we use the root mean squared error (RMSE), as well as proper scoring rules such as the negative log-likelihood (NLL) [23]. Furthermore, we also consider the mean absolute calibration error (MACE) and correlation between the predictive uncertainties and mean absolute error (MAE).[4] A detailed description of the employed metrics is in the supplementary.

# 4 Results: Examples of UQ Method Analysis

The following section provides a quantitative performance comparison of different UQ methods under a possible benchmark setting, easily enabled by our proposed framework.

## 4.1 Selective Prediction for Wind Speed Estimation

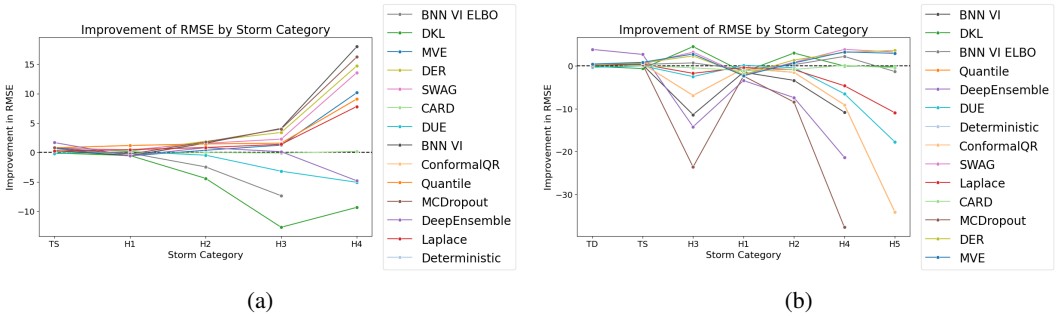

(a)             (b)

Figure 5: Selective Prediction RMSE improvement per category on the Digital Typhoon Dataset (left) and Tropical Cyclone Dataset (right).

Table 3 shows that most UQ methods improve model accuracy when applying selective prediction with respect to a deterministic baseline, which cannot express any uncertainty. Compared to Table 3, Figure 5 demonstrates a different ranking of the UQ methods, with respect to the accuracy improvement due to selective prediction, when evaluated per category, according to the Saffir-Simpson scale [66]. This ranking also differs on the DT and TC dataset, as Figure 5 shows. The skewed data distribution of both datasets, 3a and 3b, and the different uncertainty sources in the

---

[2]As available in the `timm` library [72]

[3]More examples are shown in the Github Repo for these experiments.

[4]Metrics computed with the library provided by [11]

TC and DT datasets 3.1 may contribute to these observations of aggregation pathologies. For the classification task the ranking of methods varies with Gaussian Process based methods performing better, see supplementary section 2.

Table 3: Evaluation of Regression Results on the test set. Note that [64] observe a similar ranking in terms of accuracy, also with respect to Deep Ensembles. RMSE $\Delta$ shows the improvement after selective prediction, while Coverage denotes the fraction of remaining samples that were not omitted. Selective prediction is based on the 0.8 quantile of predictive uncertainties on a validation set.

| UQ group | Method | RMSE ↓ | RMSE Δ ↑ | NLL ↓ | MACE ↓ |
|---|---|---|---|---|---|
| None | Deterministic | 9.64 | 0.00 | NaN | NaN |
| Deterministic | MVE | 10.10 | 0.64 | 3.74 | 0.06 |
| | DER | 9.59 | **1.07** | 4.32 | 0.30 |
| Quantile | QR | 9.54 | **1.03** | **3.64** | 0.05 |
| | CQR | 9.54 | **1.03** | 3.72 | 0.10 |
| Ensemble | Deep Ensemble | 14.37 | 0.77 | 4.05 | **0.01** |
| Bayesian | MC Dropout | 9.77 | **1.03** | 3.75 | 0.10 |
| | SWAG | **9.10** | 0.97 | 3.67 | 0.12 |
| | Laplace | 9.64 | 0.44 | 3.69 | 0.03 |
| | BNN VI ELBO | **9.15** | 0.17 | 15.82 | 0.35 |
| | BNN VI | 10.74 | 0.94 | 3.76 | 0.03 |
| | SNGP | 9.33 | -0.05 | 14.00 | 0.36 |
| | VBLL | 9.72 | 0.06 | 3.70 | 0.03 |
| | DKL | 10.35 | -0.31 | 3.77 | **0.01** |
| | DUE | 9.46 | -0.10 | 3.68 | **0.01** |
| Diffusion | CARD | 9.57 | 0.09 | 9.35 | 0.30 |

(a) Digital Typhoon Dataset.

| UQ group | Method | RMSE ↓ | RMSE Δ ↑ | NLL ↓ | MACE ↓ |
|---|---|---|---|---|---|
| None | Deterministic | 10.50 | 0.00 | NaN | NaN |
| Deterministic | MVE | 9.95 | 1.15 | **3.64** | 0.04 |
| | DER | 10.14 | 1.17 | 4.60 | 0.35 |
| Quantile | QR | 10.95 | **1.05** | 3.73 | **0.01** |
| | CQR | 10.95 | **1.05** | 3.79 | 0.10 |
| Ensemble | Deep Ensemble | 16.19 | **3.30** | 4.15 | 0.05 |
| Bayesian | MC Dropout | 10.23 | 0.87 | 3.81 | 0.16 |
| | SWAG | 9.78 | 1.13 | 3.71 | 0.13 |
| | Laplace | 10.53 | 0.60 | 4.31 | 0.28 |
| | BNN VI ELBO | 11.82 | 1.56 | 5.57 | 0.23 |
| | BNN VI | 11.20 | 1.45 | 3.74 | 0.02 |
| | SNGP | 12.01 | 0.28 | 5.53 | 0.18 |
| | VBLL | 10.79 | 0.51 | 3.80 | 0.07 |
| | DKL | 12.59 | 0.21 | 3.95 | 0.06 |
| | DUE | 9.95 | -0.21 | 3.73 | 0.08 |
| Diffusion | CARD | 10.86 | 0.45 | 3.92 | 0.05 |

(b) Tropical Cyclone Dataset.

Figure 6 gives a visual intuition of the selective prediction scheme. If the predictive uncertainty (red-shaded region) exceeds the established threshold (blue-shaded region), individual predictions are deferred to an expert. The models provide a reasonable mean estimate of a storm track, even though predictions are made for single image instances and the regression task is modeled by ResNet-18 without a notion of time. Figure 6 additionally showcases the effect of conformalizing the predictive uncertainty of an underlying model. Conformal prediction aims to calibrate prediction sets while providing theoretical coverage guarantees; it can be particularly interesting in the case of a high-risk task such as wind speed estimation. Figure 6b demonstrates the effectiveness of the procedure, as the coverage has increased from 0.73 to 0.97, which is also reflected in the wider prediction intervals that cover the targets without sacrificing any accuracy.

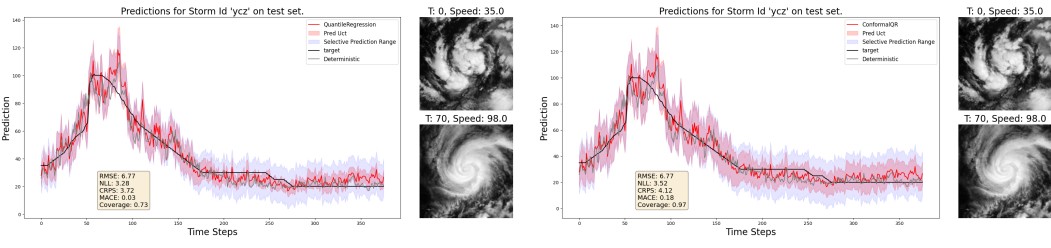

(a) Quantile Regression.  (b) Conformalized Quantile Regression.

Figure 6: Individual nowcasting predictions are stitched together to recover a time series. Areas where the red-shaded regions exceed the blue denote samples that *would* be omitted during selective prediction. The example showcases the effect of the conformal procedure, where conformalized prediction intervals increase the desired empirical coverage.

## 4.2  Photovoltaic Power Output Estimation Under Cloudy and Sunny Conditions

Figure 7 demonstrates that model performance differs under cloudy or sunny conditions. Across methods the NLL demonstrates differences in the model performance and related calibration between cloudy and sunny days. The consideration of uncertainty improved the accuracy of models compared to the deterministic baseline, as shown in the supplementary material. The correlation between the model error (in terms of MAE) and the predictive uncertainty shows a clear positive correlation (>0.45) across all methods. However, there are differences in the magnitude between methods and cloud conditions. Stakeholders might prefer good UQ estimates on more complex days, i.e., the cloudy ones, than for sunny days, where the output is much easier to predict. Exhaustive results can be found in the supplementary material.

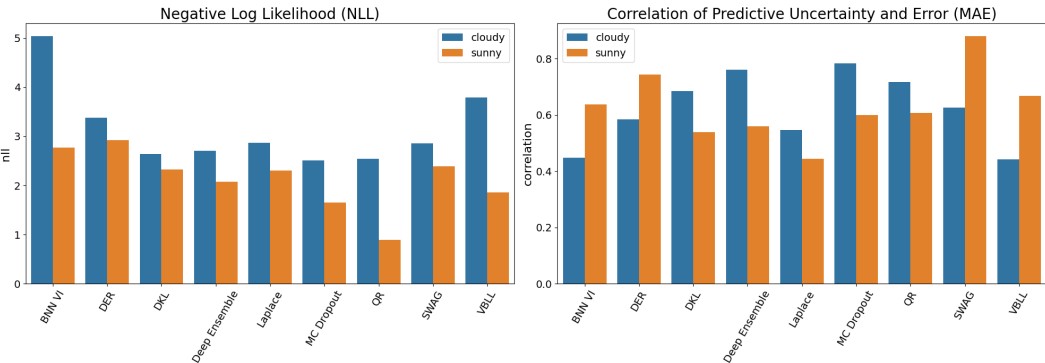

Figure 7: Negative Log Likelihood (left) and correlation between model error (measured by MAE) and predictive uncertainty for different methods on cloudy and sunny test examples.

Figure 8 showcases concrete examples with power voltage estimates plotted over the duration of a cloudy and a non-cloudy day. Compared to the smooth and consistent power output on a sunny day 8a, the predictive uncertainty is larger under cloudy conditions. This may reflect the uncertainty in the input images due to cloudiness changing faster than the time step resolution.

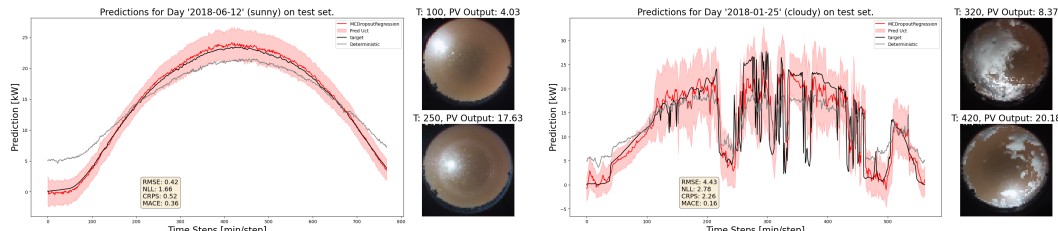

(a) MC Dropout prediction: sunny day example.    (b) MC Dropout prediction: cloudy day example.

Figure 8: Individual nowcasting predictions stitched together to recover a time series. The plot shows qualitative and quantitative differences between the two methods for the same set of predictions.

## 5 Conclusion

We have introduced `Lightning UQ Box`, a comprehensive framework for enhancing neural networks with uncertainty estimates. Additionally, we have showcased its usefulness for comparing a broad range of methods from different theoretical foundations on three relevant tasks with various sources of uncertainty. Our framework not only makes it easier for practitioners to use Bayesian methods for DL as demanded by [53] but goes beyond this by supporting UQ methods stemming from various theoretical frameworks and assumptions. Our experimental results demonstrate the differences and variability between UQ methods and, therefore, the benefit of this benchmarking framework. In conclusion, our open-source framework and the accompanying resources can be both an entry point for researchers to the field of UQ and also aid the development of new methods that address the shortcomings of existing ones [50].

## 6 Ethics and Broader Impact Statement

Including UQ in DL applied to real world and safety critical applications is of significant importance. UQ can provide the means to reduce risks, yet practitioners should not succumb to a false sense of security provided by such methods. The performance and reliability of UQ methods may be dataset and task dependent. Exactly for that reason we provide our framework under the open-source Apache-2.0 license to support open science, transparency, and collaborative research efforts.

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
