# OpenReview forum: "Lightning-UQ-Box: A Comprehensive Framework for Uncertainty Quantification in Deep Learning"
_NeurIPS.cc/2024/Datasets_and_Benchmarks_Track — Submitted to NeurIPS 2024 Track Datasets and Benchmarks_

### Official Review · Reviewer_z1fD · 2024-07-16
**Well organised UQ benchmark**

**Rating:** 7
**Confidence:** 5

**Review:**

- __Originality__: The suggestion of the two datasets seems original.
 - __Quality__: The benchmark seems of good quality. The software design seems helpful for reusability and comparison of different UQ methods.
- __Clarity__: The paper is relatively clear.
- __Significance__: The benchmark could become extremely useful to the community and with more engagement problems that I raised during my review could be addressed.

**Strengths:**

Setup of the benchmarks, preliminary test on interesting datasets.

**Additional Feedback:**

N/A

**Clarity:**

- l. 193: what is the "swath of the satellites" (I checked on Google and understood that it is "the area image on the surface", maybe a footnote or parenthesis could help here). Then I did not understand the cause/consequence of the swath of the satellite and missing pixels. Are you referring to surrounding pixels (out of the imaged area) not captured, that are missing but would in theory contain parts of the observed weather event?
- l. 202: "Ground-based fish-eye RGB images" of what? (does not seem obvious to me while looking at Figure 4.b), Figure 4.a presents Development and Test data distribution while Table 2 presents training, validation, and test. Do you mean that (1) development is training or (2) development is training and validation merged?
- l. 217: predictive uncertainty can be related to model error without necessarily being a “correlation” (that I understand as there is a linear relation between expected error and predictive uncertainty for an input $x$, expectation taken on target random variable $Y|X=x$), for example, if model bias depends on inputs. See bias-variance decomposition from Domingo et al.
- Figure 6: "The example showcases the effect of the conformal procedure, where conformal prediction intervals increase the desired empirical coverage." is this related to the level of **validity** set for the conformal prediction (the more important validity is the wider the confidence interval will be)?

**Correctness:**

- Discrepancy of results between MVE and Deep Ensembles seems strange to me as Deep Ensembles should mainly be an aggregation of MVE models. In Table 3: Deep Ensembles results seem strange, I would expect them to be on the same RMSE level as MVE. The RMSE should be about the same if the learner has no variance and smaller if the learner has variance (see Bagging paper from Breiman). Maybe NLL could be significantly worse but RMSE should not.
- Table 3: "Note that [64] observe a similar ranking in terms of accuracy, also concerning Deep Ensembles", but while opening the paper [64] and looking at Tables 4, 5, and 6 I can see that Deep Ensemble has among the best accuracy when in Table 3 it is the worst. Could you provide more explanations about this?
- Table 3: Bolding is missing in Table 3.b column RMSE and does not seem correct or consistent with column RMSE $\Delta$.
- Figure 6: "Areas where the red-shaded regions exceed the blue denote samples that would be omitted during selective prediction." visually it seems that in this example, the whole part of the extreme event when the wind speed increases would not be considered valid then, basically following back to just predicting average win speed out of extreme events... however it still looks like we are capable of predicting the event. Maybe something is not clear on how to set the threshold for selective predictions.

**Documentation:**

The benchmark is available in a well documented repository. The experiments are separated from this repository and also properly documented.

**Ethics:**

No ethical concerns

**Limitations:**

Limitations of the toolbox and proposed methods are clearly stated.

**Opportunities For Improvement:**

See my other comments about deep ensemble mainly.

**Relation To Prior Work:**

The paper performs an informative review of prior benchmark/toolbox in the field of UQ.

**Summary And Contributions:**

The paper presents a benchmark (Lightning UQ Box) for the Uncertainty Quantification of Deep Neural Networks. It focuses on two vision tasks and provides a theoretical and quantitative analysis of the implemented UQ methods. The contributions of the paper are (1) the implementation of UQ methods, (2) the modularity and usability of the code to reuse base implementations, (3) the selection of interesting tasks. The toolbox is also documented to educate about the practical and theoretical aspects of UQ. The toolbox enables the application of UQ methods on custom DL pipelines.

---

> ### Author Rebuttal · Authors · 2024-08-19
>
> Thank you for your positive feedback and the effort you have put into this review. We first answer on your proposed main opportunity for improvements, e.g., the discrepancy in the deep ensemble results. Following, we will give individual answers to the other points.
>
> **Discrepancy in Deep Ensemble Results:** Thank you very much for pointing this out! After additional diagnostic tests, we discovered that there actually was a bug in the Deep Ensemble implementation, where checkpoints were not correctly set into evaluation mode. We have fixed this and added additional unit tests and further adjusted the paper with the correct results.
>
>
> Further on Table 3:
> The results now also align better with [64]. Deep Ensembles perform among the best methods. On the Digital Typhoon dataset SWAG is marginally better than Deep Ensembles (wrt RMSE, $\Delta$ RMSE, NLL). We adjust the caption in Tab. 3 to be less ambiguous: “Note that [64] observe a similar ranking in terms of accuracy, where on specific tasks Deep Ensembles not necessarily perform best, see [64], Sec. 6.”
>
> We further adjusted the bolding in Table 3 and clarify the caption:  "For each metric the best value is in boldface."
>
> $ $
>
> **Further Points:**
>
> - **Fig. 6: [..] Visually it seems that in this example, the whole part of the extreme event when the wind speed increases would not be considered valid then, basically following back to just predicting average wind speed out of extreme events... however it still looks like we are capable of predicting the event. Maybe something is not clear on how to set the threshold for selective predictions.**
>
> Indeed, we have not been too explicit on this. Only in Table 3 we state “Selective prediction is based on the 0.8 quantile of predictive uncertainties on a validation set.” Hence, all areas where the red area exceeds the blue area correspond to predictions where the predictive uncertainty is higher than the 0.8 quantile.
>
> We adjust l. 213ff in Sec. 3.2 “Methodological Setup” to state this more explicitly:  “In the experiments, the selective prediction threshold is the 0.8 quantile of predictive uncertainties on a validation set. This means that all predictions where the predictive uncertainty is higher than this quantile are omitted in the RMSE computation. In practice,
>
> this results in referring those predictions to domain experts or further decision-making pipelines, to assess the inputs and prediction intervals obtained by the mean $\pm$ the chosen uncertainty threshold.”
>
> Indeed, for the (cherry-picked) example in Fig. 6, the targets lie in the confidence regions obtained by (C)QR. However, this should serve as an illustration of what a practitioner can do with the provided uncertainties. For increasing wind speeds the selective prediction threshold is surpassed by predictive uncertainties, but it is left to practitioners to evaluate if specifically the 0.8 quantile is needed.
>
> - **l. 193: what is the "swath of the satellites" [..]**
>
> Exactly, the "swath of the satellites" represent the area image on the surface. In particular, as satellites pass over the Earth in a polar orbit, the Earth also rotates. This results in a diagonal trajectory across the surface of the Earth. These strips of land are called "swaths". A pixel can become NaN when it is out-of-frame, i.e.,  when a part of the image of a typhoon is located at the periphery of a satellite image such that a typhoon-centered image overlaps with the boundary of the satellite image.
>
> - **l. 202: "Ground-based fish-eye RGB images" of what? [..]**
>
> This is indeed not obvious as these images are rather uncommon. The cameras take a fish-eye image of the sky. These images allow the model to predict the amount of light transmitted through (partial) cloud cover. We will update l. 202 to "ground-based fish-eye RGB images of the sky - which allow the model to predict the amount of light transmitted through partial or complete cloud cover”.
>
> - **l. 202: Fig 4a: “Development and Test data”**
>
> The “development set” refers to the merged training and validation data. The mentioned dataset comes with a predefined “development” and “test” set but no explicit split of “development” into train and validation. We clarify this in the paper
>
> - **l. 217: predictive uncertainty can be related to model error without necessarily being a “correlation” [..], for example, if model bias depends on inputs. See bias-variance decomposition from Domingo et al.**
>
> Thank you for this comment! Indeed, it may be the case that the model bias depends on the inputs and that predictive uncertainty is related to the inputs in a similar manner. Moreover, based on the mathematical foundations of most of the implemented UQ methods, it is not guaranteed that predictive error and predictive uncertainties are correlated.
> We use selective prediction to showcase a possible testing procedure to evaluate uncertainty quantification methods. However, by no means this procedure claims to be exhaustive for assessing the quality and utility of predictive uncertainties.
> Based on your remark we rephrase l. 217 to  “If the corresponding UQ method has higher uncertainties for inaccurate predictions and if leaving out the predictions for these samples increases the overall accuracy, this potentially indicates a correlation between predictive uncertainty and model error.”
>
> - **Fig. 6: [..] Is this related to the level of validity set for the conformal prediction (the more important validity is the wider the confidence interval will be)?**
>
> Indeed, this is related to the validity guarantee, which is given under certain assumptions. The bounds of the validity guarantee depend on the error rate $\alpha$, which is a hyper-parameter, that we set to 0.1. With a higher error rate, the intervals become smaller and vice versa.

---

> > ### Comment · Reviewer_z1fD · 2024-08-19
> >
> > I am glad that my comments helped. I have reviewed your answer and I will keep my rating (7: Good paper, accept).

---

### Official Review · Reviewer_hARe · 2024-07-20
**Expansive framework for UQ methods for deep learning**

**Rating:** 5
**Confidence:** 3
**Correctness:** The experimental design seems sound.
**Clarity:** The paper and code are both clear and…

**Review:**

Looking at the code this seems to be a high quality piece of work that is actively being used and developed. The number of implemented methods is quite expansive. The experimental results look reasonable, however its difficult to be sure as the application domains are fairly niche. It's difficult to say if the methods have been implemented correctly and efficiently with any comparisons to prior reference implementations of the methods. The paper would be stronger if the methods were applied on a simpler, well-known dataset and compared against prior implementations of the same methods.

**Strengths:**

The set of implemented of methods is expansive and the code is high quality.

**Additional Feedback:**

n/a

**Documentation:**

The benchmark code is available and well documented.

**Ethics:**

There is not major ethical concerns with this work.

**Limitations:**

The authors discuss the limitation that their work is built on top of PyTorch and Lightning. This is a reasonable limitation as both frameworks are widely used.

**Opportunities For Improvement:**

As discussed above it would help to have a direct comparison against previous implementation of the same methods on standard datasets.

**Relation To Prior Work:**

The authors discuss the relation to prior work to show that their framework covers more methods that currently available benchmarks.  However, they do not directly compare against them. It would have been good to see a comparison against available reference implementations of the same methods. In isolation its hard to judge how well they work and their relative runtime efficiency.

**Summary And Contributions:**

This paper provides a framework for applying deep Bayesian techniques built on top of the PyTorch and Lightning frameworks. It provides a reference implementation of a variety of Bayesian techniques under a unified API. This is motivated by the fact that UQ is important for modern networks, but is typically difficult and costly to implement. This framework provides a greater number of implementations that any previous benchmarks. They validate their methods on two use cases: cyclone wind speed prediction and photovoltaic power output estimation.

---

> ### Author Rebuttal · Authors · 2024-08-19
>
> Thank you for the effort you put in the review of the paper and the toolbox. We highly appreciate your feedback on the opportunity to improve the quality of the work by more comparisons to existing implementations and also on more common data sets.
>
> We thank the reviewer for raising this point and would like to give a response in two parts, namely additional empirical comparisons (first part) and some remarks to our modular implementations that aim to avoid efficiency issues and potential implementation errors (second part).
>
> $ $
>
> **Empirical Comparison:** As we were not exactly certain whether the reviewer had a particular reference in mind, we took the liberty to compare results to the paper Beyond Deep Ensembles: A Large-Scale Evaluation of Bayesian Deep Learning under Distribution Shift, [1]. In particular, as we want to focus on Computer Vision related tasks, we pick one regression and classification task, namely the WILDS poverty dataset and the popular CIFAR-10 Corrupted dataset. We have updated our experiment repository with code to run these [experiments](https://github.com/lightning-uq-box/experiments/tree/additional_experiments).
> Based on their supplementary material and code base we run our framework with suggested hyperparameters on the given dataset splits.
> The author’s code base reveals that they set a constant homoscedastic predictive uncertainty value of 0.1 across all methods. If we do the same for our implementation, we can confirm the reported results of Table 9 in the paper for both evaluation sets.
> The results show that the performance and log-likelihood for our implementations is similar to the results presented in [1]. All our paper and additional experiments are run on a single NVIDIA A100 GPU and for these additional experiments we observe similar run times as stated in [1].
>
> Results of the additional experiments can be found in the attached PDF.
>
> $ $
>
> **Code Accessibility:** Additionally, we follow an explicit implementation style that enables checking key equations. One can validate the code base by checking if key equations of UQ methods are implemented according to common standards. We base our implementation on reference papers with mathematical details also summarized in our comprehensive theory guide that is found in section 4 "UQ Methods Theory Guide" in the supplementary material.
>
> For example, concerning UQ methods that use the NLL loss, such as SWAG, Deep Ensembles and MVE the loss is implemented comparable to other libraries and works. We implement the equivalent NLL loss in [A] as [1] in [C]. However, in order to improve numerical stability we opt to predict the logarithm of the variance instead of the variance of a Gaussian, which is encoded in the loss function. With this approach we follow Kendall et. al [2], eq. (8), and aim at enhancing numerical stability during training which can occur when the label or target noise is near zero and the logarithm of values close to zero starts diverging. For BBB or BNNs with variational inference we use a similar loss in the Lightning UQ Box [E] compared to the Bayesian Torch library [F].
>
> $ $
>
> [1] Seligmann et al. (2024). Beyond Deep Ensembles: A large-scale evaluation of Bayesian deep learning under distribution shift. NeurIPS 24
>
> [2] Kendall, A., & Gal, Y. (2017). What uncertainties do we need in bayesian deep learning for computer vision?. Advances in neural information processing systems, 30
>
> [A]https://github.com/lightning-uq-box/lightning-uq-box/blob/d283f5cd5ddc63a3dfd6e8811c3312fafe96b4d8/lightning_uq_box/uq_methods/loss_functions.py#L156
>
> [C] https://github.com/Feuermagier/Beyond_Deep_Ensembles/blob/b805d6f9de0bd2e6139237827497a2cb387de11c/src/algos/util.py#L23
>
> [E] https://github.com/lightning-uq-box/lightning-uq-box/blob/d283f5cd5ddc63a3dfd6e8811c3312fafe96b4d8/lightning_uq_box/uq_methods/bnn_vi_elbo.py#L224
>
> [F] https://github.com/IntelLabs/bayesian-torch

---

> > ### Author Rebuttal · Authors · 2024-08-30
> >
> > Dear Reviewer hARe,
> >
> > Thank you again for your thoughtful feedback on our work.
> >
> > In addition to our initial rebuttal, we have conducted further experiments on the CIFAR-10 and WILDS datasets to strengthen our submission. These new results, which extend the previous experiments and those reported in the manuscript, are included in the attached PDF.
> >
> > We plan to continue this work, expand to more architectures and datasets, and provide a dedicated section on our documentation page that offers a comprehensive overview of these results alongside the necessary code and config files to reproduce these results.
> >
> > As the author-reviewer discussion period ends in two days, we would be especially grateful if you could take a moment to review these additional experiments and share any further thoughts or feedback with us. We highly appreciate any additional feedback and insights you can offer.
> >
> > Thank you again for your time and consideration!

---

### Official Review · Reviewer_H9aE · 2024-07-24
**Lightning UQ box: a unified approach to estimation with UQ**

**Rating:** 5
**Confidence:** 2
**Correctness:** Yes
**Clarity:** Yes

**Review:**

The main drawback of the paper is the lack of a novel framework to analyze uncertainty of DNN outputs. The proposed approaches investigated already exist in the literature although not as a package. Although it is a useful product for practitioners, it serves as a software product based on existing approaches.

**Strengths:**

The authors have explored the UQ of DNN output from the standpoint of a wide range of approaches including deterministic approaches, quantile models, ensembles, Bayesian methods, Gaussian process methods and conditional generative models and evaluated in context of vision specific task.

**Additional Feedback:**

NA

**Documentation:**

Yes

**Limitations:**

Limitations and potential negative societal impacts are adequately described

**Opportunities For Improvement:**

If the authors can describe a unifying umbrella of datasets, where one would expect one of the UQ methods to perform best (suggesting some metric) it could increase the impact of the paper.

**Relation To Prior Work:**

This part needs significant improvement since the approaches implemented for UQ already exist in the literature. How the UQ provided by Lightning UQ-box can be used by practitioners to decide on the go to approach for UQ is needed.

**Summary And Contributions:**

The Lightning UQ-box provides a comprehensive set of approaches to assess uncertainty in DNNs. In this connection, deterministic approaches, quantile models, ensembles, Bayesian methods, Gaussian process methods and conditional generative models have been provided.

---

> ### Author Rebuttal · Authors · 2024-08-19
>
> Thank you for the effort that was put into this revision and for the important questions that are raised. Please find our answers below.
>
>  $ $
>
> **If the authors can describe a unifying umbrella of datasets, where one would expect one of the UQ methods to perform best (suggesting some metric) it could increase the impact of the paper.**
> $ $
> Indeed, a definitive answer to which UQ method is best suited for specific datasets and downstream tasks would be a significant advancement in the field. However, such a definitive answer so far has not been established for DL UQ methods on real world applications and still remains elusive. Different UQ methods excel in different contexts and with respect to different metrics. Beyond that,  each method makes implicit trade-offs, e.g. between computational complexity, prediction accuracy and uncertainty calibration. The ideal choice may therefore depend on the specific downstream decision making task – even if the dataset is held fixed. This is for example further analyzed in “Specialized Uncertainties for Specialized Tasks” [1] and  in [2], where different behaviors of Bayesian approaches on different network architectures are shown.
>
> Another aspect is that identifying a unified approach for practitioners for choosing a UQ method is difficult, because many different metrics, scoring rules and values for assessing the performance of UQ in ML exist. For example, in Seligmann et al. [2] calibration metrics that differentiate underconfident and  overconfident model predictions by incorporating a sign into the often used calibration metrics, the quantile calibration error for regression tasks and the expected calibration error for classification tasks. Another example is the wide range of scoring rules that are presented and assessed in a comparison to cross validation, as well as an experimental study on probabilistic weather forecasts is presented in Gneiting et al. [3]. In most complex applications the practitioner needs to assess which metrics and scores are the most beneficial for the given down-stream task. For example, we use two different ways to assess the quality of UQ methods on the two different tasks. We do this to illustrate that there is not necessarily a unified way of choosing one best UQ method for all tasks.
>
> Therefore we argue, there is no way around evaluating the quality of different UQ methods for the specific dataset and decision making task at hand. This is precisely where our work makes a key contribution:  having an open-source library at hand that implements a wide range of methods with  unifying APIs  accelerates this  process and makes comparisons easy.  Importantly, the toolbox enables researchers to spend more time on this analysis than on the implementation.
>
> $ $
>
> **This part needs significant improvement since the approaches implemented for UQ already exist in the literature.**
>
> Indeed, various implementations of UQ methods already exist in research papers and in different software libraries. Many UQ implementations  are method-specific libraries implemented alongside  paper publications. In contrast our  library covers a large set of these methods under a unifying API together with  the established  processes and procedures from modern software development .  This, together with an already increasing adoption from the community, leads  to a well-maintained library and with feedback to ever increasing quality.
>
> Hence, our key contribution is an open-source library that implements a wide range of state-of-the-art methods with  unifying APIs  that  accelerate the comparison between methods for researchers and practitioners. As we demonstrate in Table 1, our toolbox supports the widest range of UQ methods while at the same time being easy to use and accessible as it is open-source.
>
> Moreover, we try to align with the scope of the Nips Benchmark Track. The scope of this track requests “[...] all work on data-centric machine learning research and open-source libraries and tools that enable or accelerate ML research [...].”
>
> In relation to prior work we thus argue that our toolbox explicitly enables to accelerate and improve ML research by providing comparable and accessible implementations of state-of-the-art UQ methods for DL. The novelty here is a new library with enhanced functionality and supported methods that are specifically adapted to computer vision problems. Further, such a framework has also been called for in the existing literature [4].
>
> $ $
>
> **How the UQ provided by Lightning UQ-box can be used by practitioners to decide on the go to approach for UQ is needed.**
>
> As discussed above obtaining a “go to approach” for which UQ method to use is highly context dependent and the context, in terms of down-stream task, risk-reward trade-off, evaluation metrics and scores is highly variable. Thus, we aim to provide researchers and practitioners with the relevant tools to establish this context in terms of, for example, providing common APIs to enable a unified framework for comparison of a wide range of UQ methods. This means that it is enough to copy paste a few example lines of code and to adjust the configuration files in order to apply UQ methods. However, choosing relevant tasks and metrics for evaluating and ranking the performance of UQ methods is a difficult analysis task that the practitioners and researchers need to perform.
>
> $ $
>
> [1] Mucsányi et al. (2024). Benchmarking Uncertainty Disentanglement: Specialized Uncertainties for Specialized Tasks. ICML WS 24
>
> [2] Seligmann et al. (2024). Beyond Deep Ensembles: A large-scale evaluation of Bayesian deep learning under distribution shift. Nips 24
>
> [3] Gneiting, T. et al. (2007). Strictly proper scoring rules, prediction, and estimation. Journal of the Am. Stat. Assoc., 102(477), 359-378
>
> [4] Papamarkou et al. (2024). Position paper: Bayesian deep learning in the age of large-scale ai. ICML 24

---

> > ### Comment · Reviewer_H9aE · 2024-08-30
> >
> > Based on the author response, I will increase a point on my overall scores.

---

### Official Review · Reviewer_mBbc · 2024-07-29
**A thorough comparison with existing works using a new UQ toolbox**

**Rating:** 7
**Confidence:** 4
**Correctness:** Yes
**Clarity:** Yes

**Review:**

- This work focuses on an important domain of UQ in deepl earning and is implemented in good quality.
- The paper is well-written
- Minor modification is needed

**Strengths:**

- The benchmark is well implemented and documented
- This paper is generally well written
- The comparison with existing works is thorough

**Additional Feedback:**

No additional feedback

**Documentation:**

Yes

**Limitations:**

- I only have one suggestion on Table 1: the author may consider whether it is better to put method types to a "Category" column, as opposed to use the same column as Publication.

**Opportunities For Improvement:**

- I only have one suggestion on Table 1: the author may consider whether it is better to put method types to a "Category" column, as opposed to use the same column as Publication.

**Relation To Prior Work:**

Yes

**Summary And Contributions:**

- This work presents a comprehensive framework for uncertainty quantification in deep learning.
- Two vision tasks are used to demonstrate the utility of the toolbox.

---

> ### Author Rebuttal · Authors · 2024-08-19
>
> Thank you for this valuable feedback. We agree that the previous table organization was a bit confusing. We split the categories into a separate column and moved the "Publication" row header to a column header. You can find the adjusted table in the attached PDF.

---

### Author Rebuttal · Authors · 2024-08-19

We would like to thank all reviewers for the time and effort that was put into the reviews. We highly appreciate the feedback, the raised questions, and the suggestions for further improvements on the presentation and evaluation of the provided Lightning-UQ-Box. We especially appreciate the comments that reflect the effort we have put into the accessibility of the framework (“a high quality piece of work”, “actively being used and developed”, “could become extremely useful to the community”).

The two key criticisms regarding our work are significance and validity. Regarding significance there is a documented need for accessible tool boxes:

*“Software development is key to encouraging deep learning practitioners to use Bayesian methods. More generally, there is a need for software that would make it easier for practitioners to try BDL in their projects.”* [1]

This is in line with the scope of the datasets and benchmark track that requests *“[...] all work on data-centric machine learning research and open-source libraries and tools that enable or accelerate ML research [...].”* Our toolbox answers this call by providing an open-source library with a large variety of UQ methods under a unifying API that follow modern software practices.

The second point is technical validity. In this rebuttal we provide experiments on common datasets for a direct comparison of the intersection of UQ method implementations that were also used in [2]. The results can be found in the attached PDF - for more details we would like to refer to our answer to reviewer “hARe”. This shows that our implementations of UQ methods perform comparably with equivalent libraries and implementations from the literature, while obtaining similar computational complexity.

Thank you again for your effort and please refer to our answers to the individual reviewers for all other points raised within the individual reviews.

$ $

[1] Papamarkou et al. (2024). Position paper: Bayesian deep learning in the age of large-scale ai. ICML 24

[2] Seligmann et al. (2024). Beyond Deep Ensembles: A large-scale evaluation of Bayesian deep learning under distribution shift. NeurIPS 24

---

### Decision · Program_Chairs · 2024-09-26

**Decision:**

Reject

**Comment:**

The paper provides a set of approaches to assess uncertainty in DNNs (including deterministic approaches, quantile models, ensembles, Bayesian methods, Gaussian process methods and conditional generative models).  There are two use cases, cyclone wind speed prediction and photovoltaic power output estimation, and there is a broad range of evaluations.  There is not a good discussion and evaluation of prior work.  UQ is a huge area, and the paper is certainly not a comprehensive evaluation, so a more targeted evaluation would  help the paper.  One of the most notable features of DNNs is their overparamaterization, and many statistical methods for UQ break down in that regime, and there is not an evaluation of this, which limits the usefulness as a general UQ benchmark for DNNs.